# Exploring NCATS in-house biomedical data for evidence-based drug repurposing

**Fang Liu[1], Andrew Patt[2], Chloe Chen[1], Ruili Huang[2], Yanji Xu[1], Ewy A. Mathé[2], Qian Zhu[2]***

1 Division of Rare Diseases Research Innovation, National Center for Advancing Translational Sciences (NCATS), National Institutes of Health (NIH), Bethesda, Maryland, United States of America, 2 Division of Pre-Clinical Innovation, National Center for Advancing Translational Sciences (NCATS), National Institutes of Health (NIH), Rockville, Maryland, United States of America

* qian.zhu@nih.gov

## Abstract

Drug repurposing is a strategy for identifying new uses of approved or investigational drugs that are outside the scope of the original medical indication. Even though many repurposed drugs have been found serendipitously in the past, the increasing availability of large volumes of biomedical data has enabled more systemic, data-driven approaches for drug candidate identification. At National Center of Advancing Translational Sciences (NCATS), we invent new methods to generate new data and information publicly available to spur innovation and scientific discovery. In this study, we aimed to explore and demonstrate biomedical data generated and collected via two NCATS research programs, the Toxicology in the 21st Century program (Tox21) and the Biomedical Data Translator (Translator) for the application of drug repurposing. These two programs provide complementary types of biomedical data from uncovering underlying biological mechanisms with bioassay screening data from Tox21 for chemical clustering, to enrich clustered chemicals with scientific evidence mined from the Translator towards drug repurposing. 129 chemical clusters have been generated and three of them have been further investigated for drug repurposing candidate identification, which is detailed as case studies.

## Introduction

Drug discovery is an expensive area of research and development in terms of both time and financial resources. The time frame for developing new treatments can range from 3 to 20 years and the associated costs can reach tens of billions of dollars [1]. Drug repurposing is a strategy for identifying new uses for approved or investigational drugs that are outside the scope of the original medical indication [2]. Even though many repurposed drugs have been found serendipitously in the past [3, 4], more systemic and data-driven approaches for drug candidate identification are becoming increasingly prominent. Given advancements in computational technology and science, the amount of biomedical data has recently exploded, thereby offering tremendous opportunities for supporting drug repurposing, from the design of clinical studies to improving understanding of how to target molecular mechanisms to modulate disease processes. With the mission of National Center of Advancing Translational

**Data Availability Statement:** All relevant data are within the paper and its Supporting information files.

**Funding:** This project was supported by the intramural program (ZIC TR000410-05) at NCATS,

and there was no additional external funding received for this study.

**Competing interests:** The authors have declared that no competing interests exist.

Sciences (NCATS), turning research observations into health solutions through translational science, diverse types of biomedical data have been generated and accumulated in the past decade through multiple biomedical programs and initiatives managed by the NCATS. The effort includes the Toxicology in the 21st Century program (Tox21) [5] and NCATS Biomedical Data Translator (Translator) [6], which provides complementary types of data from bioassay screening data to pathophysiology (i.e., the study of abnormal changes in body functions that are the causes, consequences, or concomitants of disease processes [7]) related data including objective signs and symptoms of disease, drug effects, and intervening types of biological data, has been selected and applied in this study. Tox21 established a library of around 10,000 compounds, containing roughly 3,700 approved and investigational drugs and 5,200 environmental chemicals [8]. The Tox21 library has been screened against over 70 in-vitro assays (e.g., assays to identify compounds that interfere with nuclear receptor signaling or stress response pathways). All data and detailed assay descriptions with target annotations are publicly available (https://tripod.nih.gov/tox/pubdata/) and PubChem database [9]. Most of these assays cover targets/pathways related to nuclear receptor signaling (NR, 55.90%), stress response (SR, 11.80%), cytotoxicity (8.80%), and other toxicity-related targets/pathways (23.50%). Data from Tox21 has been systematically preprocessed and performed quality control (QC, verifying the data quality) for toxicology applications [10–12], thereby providing a valuable source as biological activity data and can therefore be used for drug repurposing. Biomedical Data Translator ("Translator") is a multi-institution effort to develop a distributed computational reasoning and knowledge exploration system [6]. Translator has integrated over 250 knowledge sources, including highly curated biomedical databases such as Comparative Toxicogenomics Database (CTD) [13], ontologies such as Mondo, the Monarch Disease Ontology [14], and multiple NCATS owned resources, i.e., Genetic And Rare Diseases Information Center (GARD) [15], Pharos [16]. With heterogenous types of biomedical data and reasoning mechanisms implemented within Translator, it is thus a valuable resource of scientific evidence to be explored for supporting various types of biomedical applications [17, 18], including drug repurposing [19].

Prominent studies have introduced and explored the use and integration of heterogeneous types of biomedical data for drug repurposing applications. Santamaría et al developed DISNET, a knowledge base with a large complex network that stores information about diseases, symptoms, genes, and drugs extracted from different public sources [20]. DISNET has been applied to uncover novel patterns and associations and leads to hypotheses for new drug repurposing case studies [21], including COVID-19 [22]. Peyvandipour et al introduced a systems biology approach for drug repurposing by building a drug-disease network with all interactions between drug targets and disease-related genes in the context of all known signaling pathways [23]. Gao et al introduced KG-Predict, a knowledge graph of more than one million associations for 61 thousand entities from various genotypic and phenotypic databases, for drug repurposing [24]. Zeng et al [25] constructed a biomedical knowledge graph with main types of data from various resources including DrugBank, Supertarget, etc. for supporting drug repurposing. Zhu et al developed an integrative knowledge graph named NCATS GARD Knowledge Graph (NGKG), with rare diseases from GARD as a backbone and various rare disease related resources [15]. The Board Drug Repurposing Hub (BDRH) was aimed at manual curating a collection of 4,704 compounds, experimentally confirming their identifies, and annotating them with literature-reported targets [26]. The Illuminating the Druggable Genome (IDG) program has collected and organized information about protein targets, representing the most common druggable targets with an emphasis on understudied proteins. IDG manages two resources including the Target Central Resource Database (TCRD) collating heterogeneous gene/protein datasets and Pharos [16] providing interfaces to access data from

TCRD [16]. In this study, we explored the BDRH and Pharos to obtain chemical/drug and disease associations, and applied the NGKG along with data from Translator to validate drug repurposing results. Meanwhile, the advanced computational techniques, like machine learning, deep learning has been actively applied to learn patterns in biomedical data related to drugs and then link them to support the discovery of alternative uses of drugs [27–29]. We clustered Tox21 chemical compounds by using the Self Organizing Map (SOM) [30] and hierarchical clustering algorithm [31], which laid out the foundation of drug candidate identification from those clusters of chemicals.

In this study, we used bioassay screening data from Tox21 to identify clusters of drugs with similar biological activities for novel drug repurposing candidate discovery, then we explored data from the NGKG and Translator to identify direct or indirect scientific evidence for validation. More specifically, we present stepwise methods for candidate discovery, including chemical compound clustering, gene annotations for clustered chemicals and gene enrichment analysis for enriched gene identification for each cluster, from where we were able to find novel genes to each cluster in the Methods section; then followed by case studies to prove the novel genes identified from the above steps and infer new associations to diseases via the identified genes by exploring biomedical data from the NGKG and Translator.

## Methods & materials

In this study, we utilized bioassay screening data from Tox21 to identify drug repurposing candidates and validated them with scientific evidence mined from the Translator ecosystem and the NGKG. The overview of the method is shown in Fig 1.

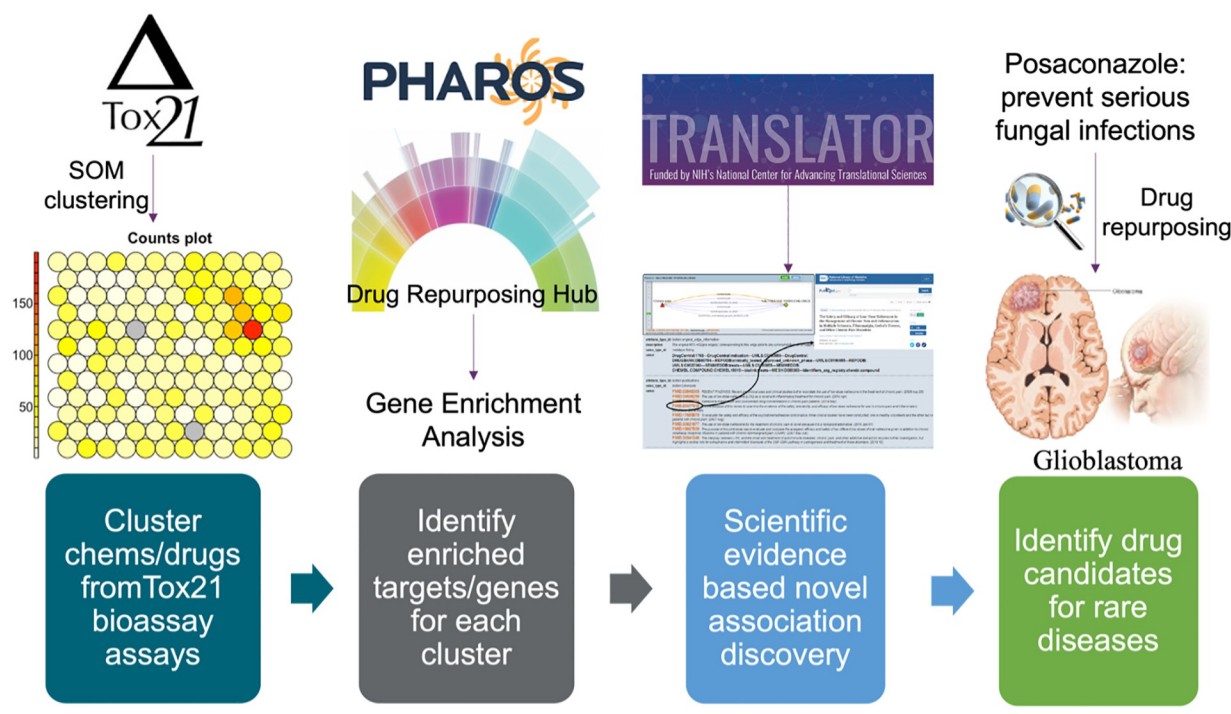

**Fig 1. Overview of the drug repurposing framework.**

**Table 1. Examples of in-vitro bioassays used in the Tox21 program.**

| qHTS Assay | Assay Target |
|---|---|
| tox21-ahr-p1 | Identifies small molecule that activate the aryl hydrocarbon receptor (AhR) signaling pathway |
| tox21-ap1-agonist-p1 | Identifies small molecule agonists of the AP-1 signaling pathway |
| tox21-ar-bla-agonist-p1 | Identifies small molecule agonists of the androgen receptor (AR) signaling pathway |
| tox21-are-bla-p1 | Identifies small molecule agonists of the antioxidant response element (ARE) signaling pathway |
| tox21-car-agonist-p1 | Identifies small molecule agonists of the constitutive androstane receptor (CAR) signaling pathway |
| tox21-tshr-agonist-p1 | Identifies small molecule agonists of the thyroid stimulating hormone receptor (TSHR) signaling pathway |

## Tox21 data preparation

The Tox21 10K compound library contains ~10,000 (8,971 unique) substances, including drugs, pesticides, consumer products, food additives, industrial chemicals, cosmetics, etc. [32]. The qHTS data used in this analysis was generated by screening the Tox21 10K library against 78 in vitro assays (examples of bioassays are given in Table 1 and a complete list can be found on the public Tox21 website [33]). Compound activity scores are reported using the curve rank metric, which is valued between -9 and 9 determined by several features of the primary concentration-response curve including potency, efficacy, and quality. A large positive curve rank represents strong activation while a large negative curve rank represents strong inhibition of the assay target. Of the 8,971 substances in the original dataset, 7,170 had curve rank data across all the Tox21 in-vitro bioassays and only those compounds with activity data were used.

## Tox21 compound clustering

We hypothesized that compounds with similar biological activity profiles may share similar targets or modes of action. We clustered 7,170 compounds in the Tox21 10K library based on their bioassay screening data by applying the self-organizing map (SOM) model, which has been proved useful to model the Tox21 10K chemical profiles for in vivo toxicity prediction and mechanism characterization [10]. Specifically, we fit a SOM model with the bioassay data as input using the *Kohonen* package in R, [34] and a pairwise Euclidean distance metric.

Because the numbers of compounds within the SOM clusters were not equally distributed, which could negatively impact the subsequent gene enrichment analysis, we merged small SOM clusters with the number of compounds less than fifteen, using hierarchical clustering of the SOM centroids. The hierarchical clustering was performed using the "complete" agglomeration method based on Pearson correlation coefficients between SOM cluster centroids. This approach merged small SOM clusters with adjacent SOM clusters that showed highest similarity.

## Identifying gene targets enriched in each cluster

**Collecting gene annotations.** To collect known gene targets for 7,170 Tox21 chemicals, we harnessed publicly available associations between chemicals and genes from Pharos [16] and the Board Drug Repurposing Hub (BDRH) [26]. Pharos and the BDRH provide comprehensive and complementary chemical and gene associations, which describes in the Results section. We first mapped Tox21 chemicals to Pharos and the BDRH based on InchIKeys

**Table 2. A contingency table for gene target enrichment analysis.**

|  | Compounds targeting the gene | Compounds targeting other genes |
|---|---|---|
| Within the cluster | a | b |
| Outside the cluster | c | d |

a: number of compounds targeting the gene within the cluster.

b: number of compounds targeting the gene outside the cluster.

c: number of compounds targeting other genes within the cluster.

d: number of compounds targeting other genes outside the cluster.

which were converted from SMILES generated for each Tox21 chemicals with RDKit [35]. Notably, only the main component with the longest SMILES string in each compound structure was applied for InChIKey conversion and the first 14 characters in the InChIKey as the primary key was used for chemical mapping. This step ensured that salts and stereo chemistry were removed for chemical mappings. Once the chemicals mapped, we retrieved gene annotations for those mapped chemicals from Pharos and the BDRH.

**Gene target enrichment analysis and pathway enrichment analysis.** After obtaining the associated gene target(s) for chemicals from the above step, we performed gene target enrichment analysis to identify gene targets enriched in each cluster. A contingency table was created to calculate gene frequency inside or outside a certain cluster (see Table 2 for the gene target enrichment use case). Significance of gene enrichment in a cluster was evaluated using one-tailed Fisher's exact test [36], followed by multiple testing corrections with the Bum class implemented in Bioconductor/ClassComparison [37]. In the following analyses, we selected enriched genes in a cluster using a false discovery rate (FDR) cutoff of 1%.

**Evidence based drug repurposing.** The Translator leverages integrated data from over 250 knowledge sources including highly curated biomedical data and derived clinical data [6], which represents various types of data, such as Disease, SmallMolecule, ClinicalFinding, Cell, etc. and the corresponding relationships including treats, gene_associated_with_condition, has_phenotype, has_target in Biolink model [38]. Given such big biomedical data integrated and presented in KGs within the Translator, it illustrates great opportunities to support evidence based drug repurposing. More specifically, the enriched genes were identified for each cluster, thus we aimed at identifying novel associations among enriched genes and chemicals and possible related diseases by accessing the Translator, particularly ARAX [39], a Translator tool. We selected three clusters for discovering potential drug repurposing candidates, which describes in case studies.

## Results

### Clustering results

Chemicals from the Tox21 library were grouped into 142 clusters based on their bioassay activity profile similarity (i.e., the curve ranks) using the SOM algorithm. The complete clustering results can be found in the S2 Data. The SOM clustering results are shown in Fig 2. Clusters with more chemicals shown in dark yellow or red dots in the counts plot, are nearly inactive against most of the bioassays. The distribution of clusters based on the number of compounds is shown in Fig 3, where we can find that most clusters are associated with a small number of compounds, less than 50. Thus, we further merged the small clusters based on hierarchical clustering (see Methods).

## Counts plot

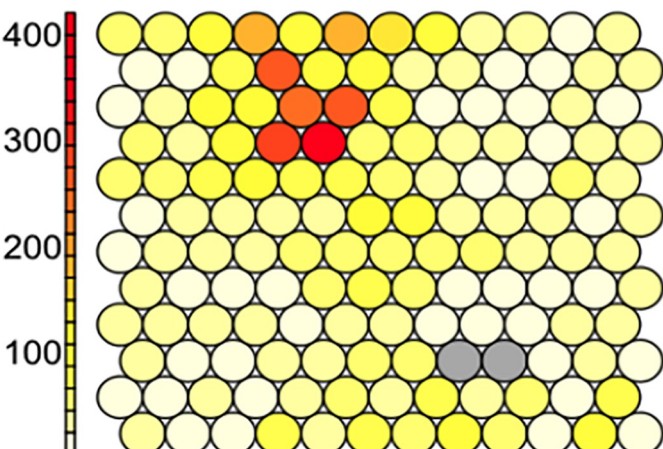

**Fig 2. SOM clustering results.** The dot denotes a cluster of chemicals, and the color of dots corresponds to the size of the clusters (clusters with more chemicals shown in dark yellow or red dots).

After hierarchical clustering applied over the SOM clusters, we merged 24 highly correlated SOM clusters with less than 15 compounds based on Pearson Correlation Coefficient. For example, we merged the cluster #117 with cluster #105 via hierarchical clustering. We retained the cluster number 105 since #105 containing more compounds than #117. After merging, 129 clusters remained, and gene enrichment analysis was then performed on these clusters. The complete clustering results can be found in the S1 Data.

To validate the performance of clustering algorithms, we examined chemical similarity among those clusters. We obtained an average Tanimoto coefficient of 0.099 for more than 24

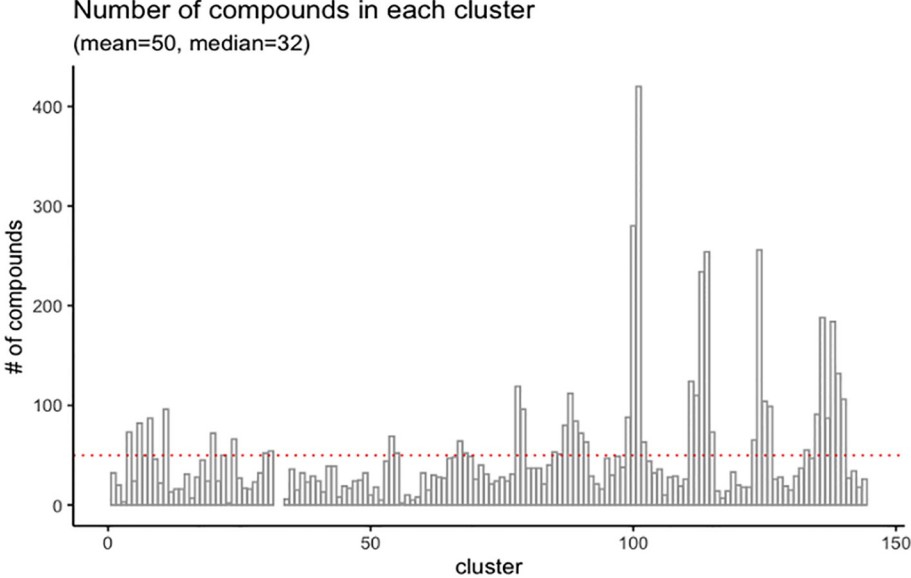

**Fig 3. Distribution of the SOM clusters based on the number of compounds.**

million unique chemical pairs across all clusters, and the average is almost doubled when we looked at the intra-cluster coefficient of 0.171. Although the overall Tanimoto coefficient is low given the diversity of Tox21 chemical compounds, it indicates those chemicals within the clusters are more structurally similar than between clusters.

## Gene target enrichment and pathway enrichment analysis

Of the 7,170 chemical compounds with bioassay data, we generated SMILES for 7,030 compounds and the corresponding InChiKeys for 7,017 compounds. We identified a total of 1,001 unique genes that could target 1,535 compounds from Pharos, and 1,303 unique genes for 1,346 compounds from the BDRH. By combining these two sets, we mapped 1,829 distinct compounds associated with 1,629 unique genes. 1,318 or 72% of these 1,829 compounds are FDA approved drugs, 600 are procured from the EPA, and 470 are procured from the NTP. Fig 4 shows overlaps of genes (Fig 4a) and chemical compounds (Fig 4b) from Pharos and the BDRH. Clearly more gene targets were obtained from the BDRH than Pharos (Fig 4a), and more compounds from Pharos than the BDRH (Fig 4b). The complete compound and gene relationships can be found in the supplemental materials.

Once we obtained associated genes for chemicals from each cluster, we performed enrichment analysis against the 129 clusters, testing the overrepresentation of gene target associations with compounds present in each cluster. Of those 129 clusters, 120 clusters had one or more enriched genes based on the p-value cutoff value of 0.0086, as calculated by the Bum class (see Methods). The number of enriched gene targets for each cluster varies from 1 to 65, with a mean of eight targets. Fig 5 shows the distribution of the number of enriched genes across drug clusters.

We then analyzed pathways associated with these enriched gene targets. To establish a global trend of enrichment of biological pathways within clusters, we compared our results to a pathway enrichment analysis of random drug targets grouped within clusters of the same size of the actual data. We found a much larger number of enriched pathways in the actual data than in the randomized data, confirming that compounds targeting similar pathways are clustered by our method (Fig 6).

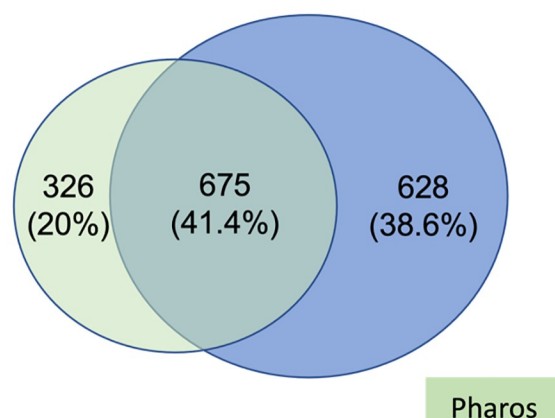
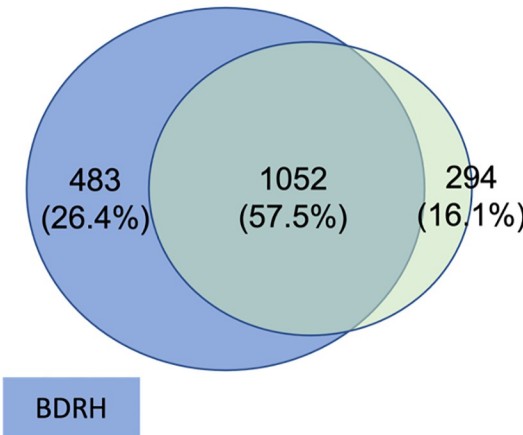

**Fig 4. Overlap in genes and compounds between Pharos and the BDRH.** a) more gene targets were found in the BDRH than via Pharos; b) Pharos had more compounds than BDRH.

## Number of enriched gene targets for each cluster

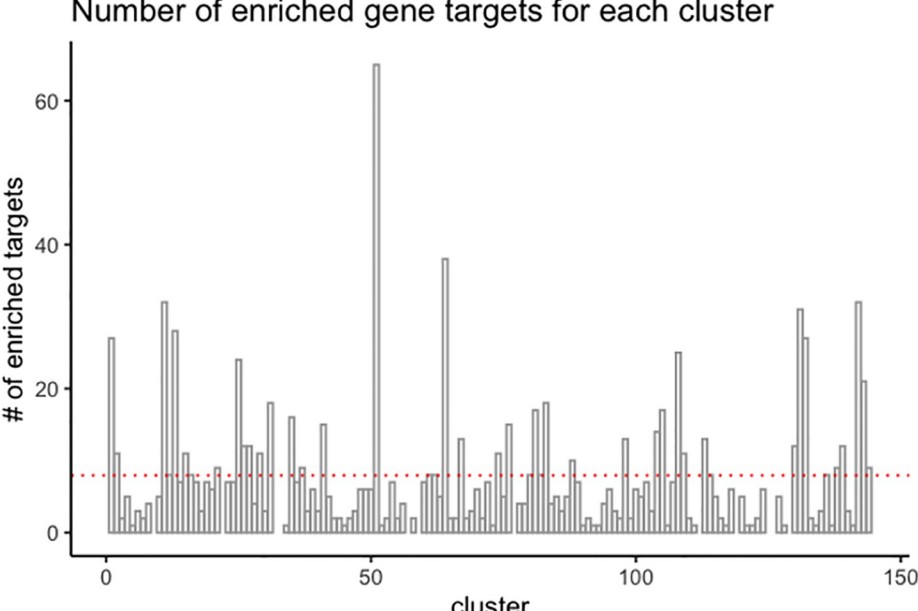

**Fig 5. Distribution of the number of enriched gene targets for each cluster.**

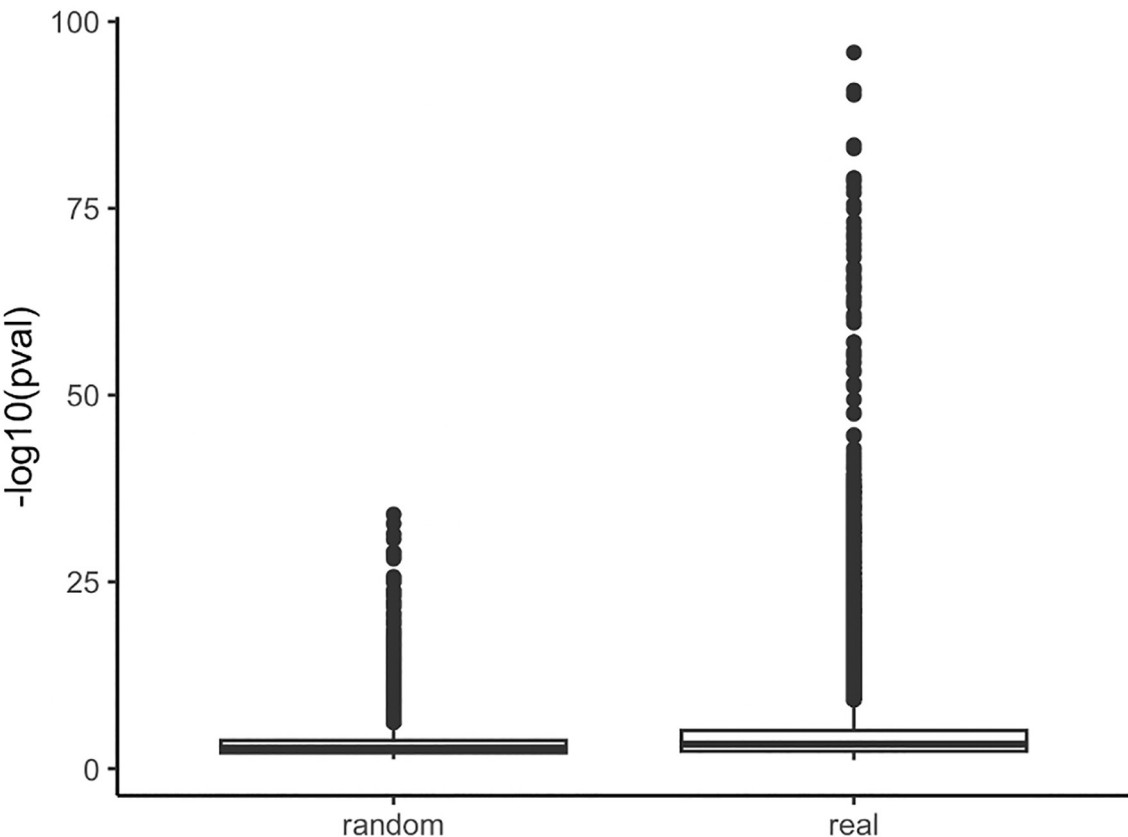

**Fig 6. Comparing pathway analysis p values in randomized gene target clusters (left) versus pathway analysis p values from the actual Tox21 drug clusters (right).**

## Drug repurposing candidate identification

We validated clusters of drugs from the above steps with evidence derived from the Translator and the NGKG in an effort to evaluate the utility of the clusters for drug repurposing. Three clusters were selected for investigation.

**Case study 1.** We found that cluster #1 is a GPCR-enriched cluster, of the 32 compounds in cluster #1, 27 compounds were associated with at least one of the enriched GPCR targets. Enriched pathways in this cluster included "Monoamine GPCRs" (Holm-adjusted p = 4.26e-75), "Amine ligand-binding receptors" (Holm-adjusted p = 3.43e-71) and "GPCRs, other" (Holm-adjusted p = 1.43e-08). G-protein-coupled receptors (GPCRs) are transmembrane proteins that reside on cell surfaces. They can detect molecules outside the cell and activate cellular responses. GPCRs are important drug targets, and about 1/3 to 1/2 of all marketed drugs act by binding to GPCRs [40].

In this case study, we aimed to validate whether GPCR gene targets in these clusters have potential associations to the compounds in cluster #1, particularly for those compounds without annotated genes identified from Pharos and the BDRH. Among the five compounds without annotated genes in this cluster, three are FDA approved drugs, Fabesetron, Ftormetazine and Difeterol. We next investigated whether these drugs had potential associations with any GPCR targets by exploring Translator as well as the NGKG.

Fabesetron is a serotonin receptor antagonist that was developed for chemotherapy-induced emesis in the 2000s, but clinical development was terminated in phase II due to reported side-effects [41]. As a member of GPCR family, HTR4 is related to Fabesetron was identified via Translator. Furthermore, additional GPCR genes were found via inference by adding one intermediate node (a wild node) between Fabesetron and GPCR genes as a query graph. Ftormetazine is a derivative of the phenothiazine class of antipsychotic drugs that act on the muscarinic cholinergic system; it is associated with Selective Serotonin Reuptake Inhibitors (SSRIs), and is a SSRI related antidepressant, which has been approved by querying the NGKG. Lastly, we found that Difeterol, an antihistamine used as an OTC drug in Japan (https://www.genome.jp/entry/D09748), is a subclass of Histamine-1 Receptor Antagonist via Translator. Details about those findings are listed in Fig 7. Collectively, these findings provide further support for cluster #1 as being primarily comprised of drugs related to GPCR-targeting that could be repurposed for diseases that involve GPCR targets.

**Case study 2.** Cluster #2 is enriched with kinase targeting compounds. Enriched pathways associated with cluster 2 include "Signaling by ERBB2 in Cancer" (Holm-adjusted p = 0.045), "PI3K events in ERBB2 signaling" (Holm-adjusted p = 0.013), and "GRB2 events in ERBB2 signaling" (Holm-adjusted p = 0.013). According to OMIM [42] and Orphanet [43], one gene among eleven enriched genes in this cluster, ERBB2 and associated pathways are linked with a wide range of cancers, including lung adenocarcinoma, gastric cancer, glioblastoma, and ovarian cancer. We first attempted to identify potential associations between ERBB2 and the compounds in the cluster via Translator. Out of nineteen compounds in this cluster, Posaconazole (PubChem:468595) is an antifungal, and can treat or prevent fungal infections, especially in people with weak immune systems. Further, we found associations between ERBB2 and Posaconazole through different intermediate drug nodes, which present drug-drug interaction with Posaconazole, shown in Fig 8.

Given the associations between Posaconazole and ERBB2 (Fig 8), and ERBB2 and glioblastoma (from OMIM), we hypothesized that Posaconazole might be repurposed for glioblastoma, which was further supported by the Translator, shown in Fig 9. Concomitantly, numerous studies have suggested there are strong relationships from azoles such as Posaconazole as a potential treatment option for glioblastoma [44–46], although the mechanisms by

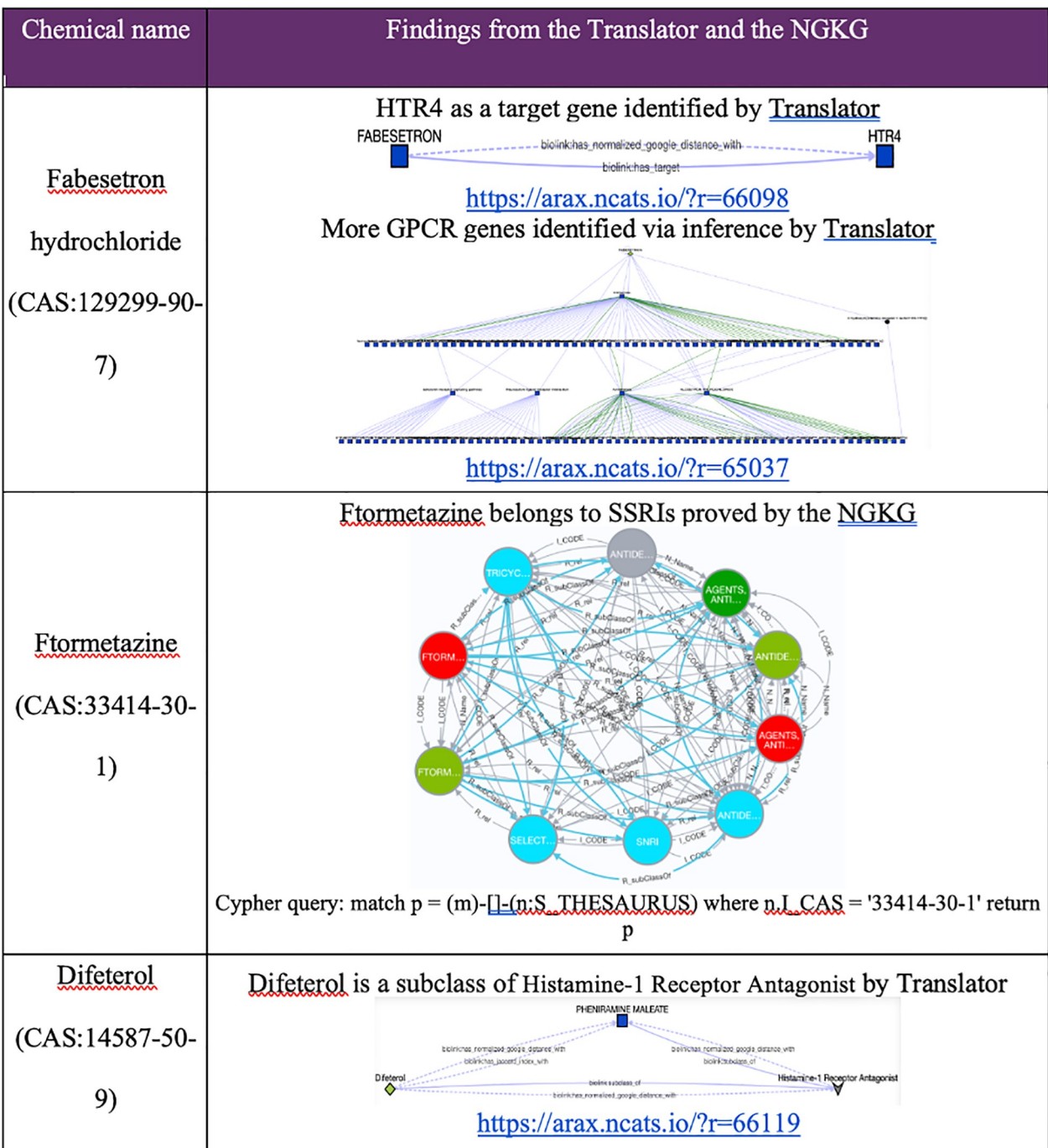

| Chemical name | Findings from the Translator and the NGKG |
|---|---|
| Fabesetron hydrochloride (CAS:129299-90-7) | HTR4 as a target gene identified by Translator<br>FABESETRON — biolink:has_normalized_google_distance_with / biolink:has_target — HTR4<br>https://arax.ncats.io/?r=66098<br>More GPCR genes identified via inference by Translator<br>https://arax.ncats.io/?r=65037 |
| Ftormetazine (CAS:33414-30-1) | Ftormetazine belongs to SSRIs proved by the NGKG<br>Cypher query: match p = (m)-[]-(n:S_THESAURUS) where n.L_CAS = '33414-30-1' return p |
| Difeterol (CAS:14587-50-9) | Difeterol is a subclass of Histamine-1 Receptor Antagonist by Translator<br>https://arax.ncats.io/?r=66119 |

**Fig 7. Gene-compound association discovery for the GPCR enriched cluster #1.**

which azoles inhibit glioblastoma cell growth have yet to be elucidated. The impact of Posaconazole on glioblastoma tumor survival in both in vivo and in vitro studies, combined with its status as a previously approved anti-fungal treatment, have led to a phase 0 clinical trial test of Posaconazole in glioblastoma [47].

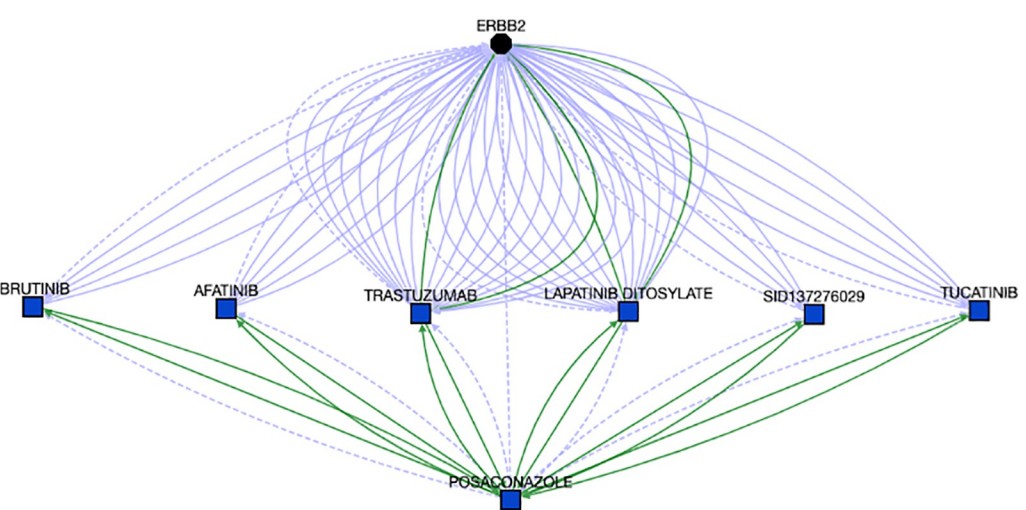

**Fig 8. Associations between ERBB2 and Posaconazole.** The details can be found at https://arax.ncats.io/?r=66179.

**Case study 3.** For cluster #105, we scanned through 25 compounds without gene annotations out of 36 compounds, to identify any potential associations between those compounds and enriched genes. One of these compounds, Kaempferol, which is a chemical found in fruits and vegetables and might reduce cancer risks and development [48], presents strong associations with DPP4, 1 of 17 enriched genes by querying Translator (Fig 10). Meanwhile we found 319 DPP4 correlated diseases, including COVID-19 (see the resulting graph at https://arax. ncats.io/?r=65921). Furthermore, we looked for inferred paths linking Kaempferol to any diseases via DPP4 and another gene target based on the route of "Kaempferol-gene-DPP4-Disease". Search results (accessible at https://arax.ncats.io/?r=65933) highlight the association between Middle East respiratory syndrome and DPP4 [49]. By synthesizing the above identified findings/associations, we concluded that Kaempferol might be used for the treatment of COVID-19. Supporting our hypothesis, Kaempferol has been reported to show anti-SARS-CoV-2 activity in vitro [50–52].

## Discussion

In this study, we demonstrated the use of NCATS in-house biomedical data for generating relevant hypotheses towards drug repurposing. Tox21 applies standard protocols to manage 10K

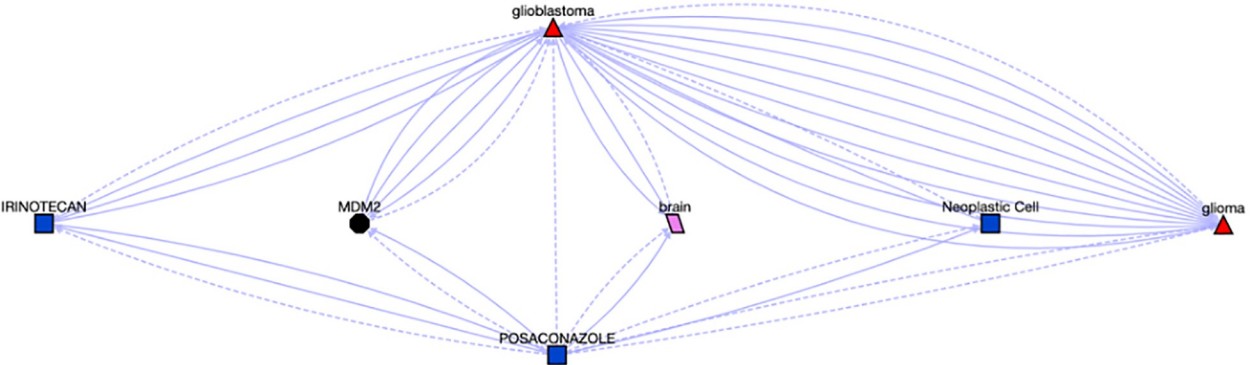

**Fig 9. Associations between Posaconazole and Glioblastoma, the details can be found at https://arax.ncats.io/?r=116621.**

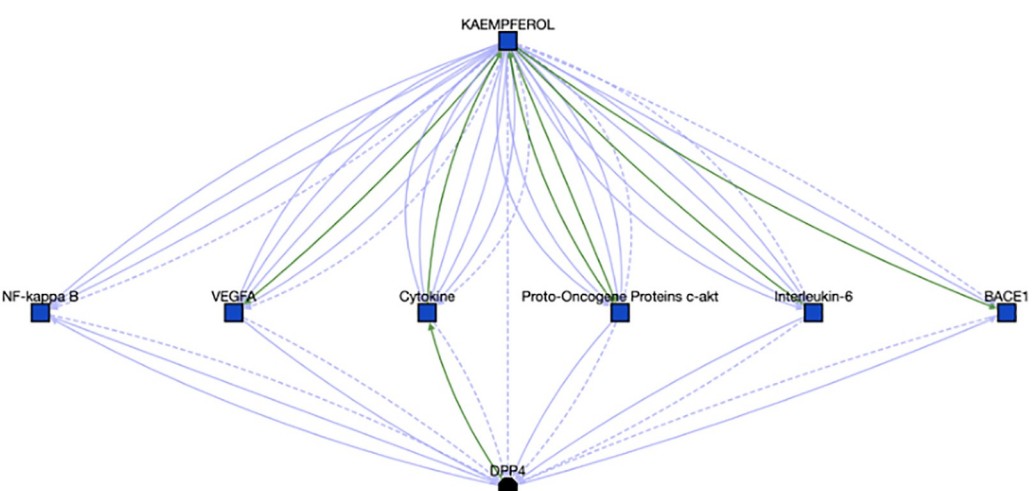

**Fig 10. Associations between Kaempferol and DPP4, the details can be found at https://arax.ncats.io/?r=65916.**

compounds of which 3,700 FDA approved and investigational drugs across 70 different bioassays and produced a robust set of screening data for toxicology applications. Translator aggregates diverse biomedical resources and inference engines for supporting various biomedical applications. Pharos provides facile access to various types of data surrounding any targets. The NGKG integrates comprehensive biomedical data pertinent to GARD rare diseases. Each of these resources provide complementary information to supplement different aspects of the present drug repurposing pipeline. We clustered Tox21 compounds based on their in vitro bioassay activity profiles uncovered underlying shared molecular mechanisms that provide key information to identify repurposed drug candidates. Pharos was applied to identify associated gene targets for Tox21 chemical compounds. We explored the Translator and the NGKG to identify scientific evidence for validating drug repurposing candidates. Although we were able to apply those resources to find potential candidates, which are illustrated in the case studies, we acknowledged the limitations of those resources and proposed extension accordingly. One caveat regarding the Tox21 bioassays is that the targets represented by these assays are not very diverse focusing primarily on two toxicity-related areas, i.e., nuclear receptor signaling and stress response. Thus, as the next step, we will include additional bioassay data, such as, PubChem Bioassay. Translator has capability of mining its underlying aggregated data to uncover hidden biomedical insight, however the current process of uncovering hidden associations/evidence is mainly relied on manual assessment and interpretation from a great number of inferred results. We manually reviewed and filtered the meaningful associations generated by the Translator for the presented three case studies. To automate this process, we will work closely with the Translator team on result organizing and ranking. Pharos and the BDRH were applied for gene and chemical association retrieval, from where associated genes have been obtained for about 26% (1,829) Tox21 compounds. As a proof-of-principle study, we did not extend the mapped genes with additional resources since our goal was to demonstrate feasibility of the pipeline for supporting drug repurposing. In the future, we will include more resources to expand the annotated gene list for Tox21 compounds to enhance the ability of gene enrichment analysis.

Tox21 compounds were clustered using SOM supplemented with hierarchical clustering based on shared biological activities based on bioassay screening data. By performing chemical structure similarity comparison and pathway enrichment analysis, we confirmed that

chemicals are more structurally related within the clusters than outside the clusters based on their chemical structures, and compounds targeting similar pathways are clustered by our cluster method. Together, the findings confirmed that the relationships between compounds, gene target, and diseases, along with structural data, could be harnessed from existing data sources such as Tox21 and be used to inform the identification of drug repurposing candidates. Future work aims to identify the biochemical and structural properties exhibited by these compounds as features to construct predictive models that can potentially evaluate a given compound's level of association to a rare disease.

We performed three case studies to demonstrate the capability of our pipeline for drug repurposing by utilizing NCATS in-house data. We identified the compounds in cluster #1 are GPCR-targeting which has been proved with scientific evidence identified from the Translator. The drugs in this cluster can potentially be repurposed for diseases that involve GPCR targets. We also found that Posaconazole, an antifungal drug might be repurposed for glioblastoma, which is in phase 0 clinical trial; and Kaempferol, a natural flavanol might be used for COVID-19. As a proof-of-concept, only three clusters were selected for investigation, as a next step, we will study more clusters from the rest of 126 clusters with consultation of subject matter experts (SMEs). All those findings can serve as initial validation of our approach and will be further evaluated by conducting biological experiments, which will be planned for the next step.

## Supporting information

**S1 Data.**
(XLSX)

**S2 Data.**
(XLSX)

## Acknowledgments

The analyses described in this publication were conducted with data and/or tools accessed through the NCATS Biomedical Data Translator (https://ncats.nih.gov/translator).

## Author Contributions

**Conceptualization:** Ruili Huang, Yanji Xu, Ewy A. Mathé, Qian Zhu.

**Data curation:** Fang Liu, Chloe Chen.

**Investigation:** Qian Zhu.

**Methodology:** Fang Liu, Ruili Huang, Qian Zhu.

**Project administration:** Qian Zhu.

**Resources:** Qian Zhu.

**Supervision:** Qian Zhu.

**Validation:** Fang Liu, Andrew Patt, Qian Zhu.

**Writing – original draft:** Fang Liu, Qian Zhu.

**Writing – review & editing:** Fang Liu, Andrew Patt, Chloe Chen, Ruili Huang, Yanji Xu, Ewy A. Mathé, Qian Zhu.

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
