## [Decision Letter · Decision Letter 0]

25 Oct 2023

PONE-D-23-21475Exploring NCATS In-House Biomedical Data for Evidence-based Drug RepurposingPLOS ONE

Dear Dr. Zhu,

Thank you for submitting your manuscript to PLOS ONE. After careful consideration, we feel that it has merit but does not fully meet PLOS ONE’s publication criteria as it currently stands. Therefore, we invite you to submit a revised version of the manuscript that addresses the points raised during the review process.

This is timely and important manuscript.  Please address the reviews comments.

We look forward to receiving your revised manuscript.

Kind regards,

Robyn L Tanguay, PhD

Academic Editor

PLOS ONE

Journal Requirements:

“This project was supported by the intramural program (ZIA TR000410-03) at NCATS”

“This project was supported by the intramural program (ZIA TR000410-03) at NCATS. **:**

“This project was supported by the intramural program (ZIA TR000410-03) at NCATS”

Additional Editor Comments (if provided):

Please address the minor comments put forward by the primary reviewer.

Reviewers' comments:

Reviewer's Responses to Questions

**Comments to the Author**

1. Is the manuscript technically sound, and do the data support the conclusions?

Reviewer #1: Yes

2. Has the statistical analysis been performed appropriately and rigorously? 

Reviewer #1: Yes

3. Have the authors made all data underlying the findings in their manuscript fully available?

Reviewer #1: Yes

4. Is the manuscript presented in an intelligible fashion and written in standard English?

Reviewer #1: Yes

5. Review Comments to the Author

Reviewer #1: The authors provide methods and supporting data illustrating how large and diverse data sets can be connected and queried with deep learning and related computational tools to generate hypotheses about compound bioactivities that may be useful in novel drug discovery. The manuscript is organized and concise and the use of three examples provides good support for the utility of the approach. The work should be of interest to those considering how to make use of large volumes of publicly available data to understand bioactivity of small molecules. Some editing of grammar and increased clarity of statements would be beneficial. It may also be useful for a moderate amount of additional explanations for much of the jargon used in the paper be provided as this may be unfamiliar to many readers. Finally, it would be useful to include a brief discussion of whether the assays used in the analysis were designed to focus on a single molecular mechanism (and, hence, expect higher chemical similarity among active chemicals) or more phenotypic assays containing multiple potential molecular targets (with more chemical diversity expected).

P 2: “pathophysiology-related”

P 3: “Data from Tox21 for toxicology applications has been systematically preprocessed and quality control conducted (QC) [8-10],”

P4: IDG manages two resources including the Target Central Resource Database (TCRD)

collating manage heterogeneous gene/protein datasets and Pharos providing interfaces to access

data from TCRD.[14]

p 4: Meanwhile, the advanced computational techniques, like machine learning and

deep learning has been actively applied to learn patterns in biomedical data related to drugs and

then link them up to the potential of treating alternative diseases.[24-26]”

p 20: “In this study, we demonstrated the use of NCATS in-house biomedical data for supporting drug repurposing.” This may be an overly strong statement. I don’t think drug repurposing was actually shown; but, rather that the approach generates relevant hypotheses.

6. PLOS authors have the option to publish the peer review history of their article (what does this mean?). If published, this will include your full peer review and any attached files.

Reviewer #1: No

---

## [Author Response · Author response to Decision Letter 0]

7 Nov 2023

Reviewer #1: The authors provide methods and supporting data illustrating how large and diverse data sets can be connected and queried with deep learning and related computational tools to generate hypotheses about compound bioactivities that may be useful in novel drug discovery. The manuscript is organized and concise and the use of three examples provides good support for the utility of the approach. The work should be of interest to those considering how to make use of large volumes of publicly available data to understand bioactivity of small molecules. 

Response: Thanks so much for the positive comments about the importance and contribution of this work!

Q1: Some editing of grammar and increased clarity of statements would be beneficial. 

Response: We thoroughly edited the whole manuscript.

Q2: It may also be useful for a moderate amount of additional explanations for much of the jargon used in the paper be provided as this may be unfamiliar to many readers. 

Response: Thanks so much for the comment! We added additional explanations on those jargons.

Q3: Finally, it would be useful to include a brief discussion of whether the assays used in the analysis were designed to focus on a single molecular mechanism (and, hence, expect higher chemical similarity among active chemicals) or more phenotypic assays containing multiple potential molecular targets (with more chemical diversity expected).

Response: Thanks so much for the comment! We added additional statements to descript the assays applied in this study. “All data and detailed assay descriptions with target annotations are publicly available (https://tripod.nih.gov/tox/pubdata/) and PubChem database[1, 2]. Most of these assays cover targets/pathways related to nuclear receptor signaling (NR, 55.90%), stress response (SR, 11.80%), cytotoxicity (8.80%), and other toxicity-related targets/pathways (23.50%).

Q4: “pathophysiology-related”

Response: we added a statement to explain pathophysiology, “the study of abnormal changes in body functions that are the causes, consequences, or concomitants of disease processes”.

Q5: “Data from Tox21 for toxicology applications has been systematically preprocessed and quality control conducted (QC) [8-10],”

Response: we added a brief explanation about QC, verifying the data quality.

Q6: IDG manages two resources including the Target Central Resource Database (TCRD)

collating manage heterogeneous gene/protein datasets and Pharos providing interfaces to access data from TCRD.[14]

Response: we added a citation for Pharos to give more information about it, https://academic.oup.com/nar/article/51/D1/D1405/6851109

Q7: Meanwhile, the advanced computational techniques, like machine learning and

deep learning has been actively applied to learn patterns in biomedical data related to drugs and then link them up to the potential of treating alternative diseases.[24-26]”

Response: We rewrote this statement, “Meanwhile, the advanced computational techniques, like machine learning, deep learning has been actively applied to learn patterns in biomedical data related to drugs and then link them to support the discovery of alternative uses of drugs.”

Q8: “In this study, we demonstrated the use of NCATS in-house biomedical data for supporting drug repurposing.” This may be an overly strong statement. I don’t think drug repurposing was actually shown; but, rather that the approach generates relevant hypotheses.

Response: Thanks so much for the comment! I edited the statement accordingly, “In this study, we demonstrated the use of NCATS in-house biomedical data for generating relevant hypotheses towards drug repurposing.”

---

## [Editor Report · Decision Letter 1]

9 Nov 2023

Exploring NCATS In-House Biomedical Data for Evidence-based Drug Repurposing

PONE-D-23-21475R1

Dear Dr. Zhu,

We’re pleased to inform you that your manuscript has been judged scientifically suitable for publication and will be formally accepted for publication once it meets all outstanding technical requirements.

Kind regards,

Robyn L Tanguay, PhD

Academic Editor

PLOS ONE

Additional Editor Comments (optional):

Thank you for completely addressing the reviewer comments and for the grammatical edits of your manuscript.
---

## [Editor Report · Acceptance letter]

20 Nov 2023

PONE-D-23-21475R1 

Exploring NCATS In-House Biomedical Data for Evidence-based Drug Repurposing 

Dear Dr. Zhu:

I'm pleased to inform you that your manuscript has been deemed suitable for publication in PLOS ONE. Congratulations! Your manuscript is now with our production department. 

Kind regards, 

on behalf of

Dr. Robyn L Tanguay 

Academic Editor

PLOS ONE